# TagOOD: A Novel Approach to Out-of-Distribution Detection via Vision-Language Representations and Class Center Learning

## ABSTRACT

Multimodal fusion, leveraging data like vision and language, is rapidly gaining traction. This enriched data representation improves performance across various tasks. Existing methods for out-of-distribution (OOD) detection, a critical area where AI models encounter unseen data in real-world scenarios, rely heavily on whole-image features. These image-level features can include irrelevant information that hinders the detection of OOD samples, ultimately limiting overall performance. In this paper, we propose **TagOOD**, a novel approach for OOD detection that leverages vision-language representations to achieve label-free object feature decoupling from whole images. This decomposition enables a more focused analysis of object semantics, enhancing OOD detection performance. Subsequently, TagOOD trains a lightweight network on the extracted object features to learn representative class centers. These centers capture the central tendencies of IND object classes, minimizing the influence of irrelevant image features during OOD detection. Finally, our approach efficiently detects OOD samples by calculating distance-based metrics as OOD scores between learned centers and test samples. We conduct extensive experiments to evaluate TagOOD on several benchmark datasets and demonstrate its superior performance compared to existing OOD detection methods. This work presents a novel perspective for further exploration of multimodal information utilization in OOD detection, with potential applications across various tasks. Code will be available.

## CCS CONCEPTS

• **Computing methodologies** → *Object recognition*.

## KEYWORDS

Out-of-distribution detection, Vision-Language representations, Representative class centers

## 1 INTRODUCTION

Most modern deep neural networks [7, 14, 15, 27, 32, 47] are validated using test data from the same distribution as the training data. Nevertheless, encountering out-of-distribution (OOD) samples is inevitable when deploying deep learning models in real-world scenarios. This phenomenon highlights the importance of an ideal recognition model, which should not only deliver accurate predictions on the training distribution (referred to as in-distribution data)

**Unpublished working draft. Not for distribution.**

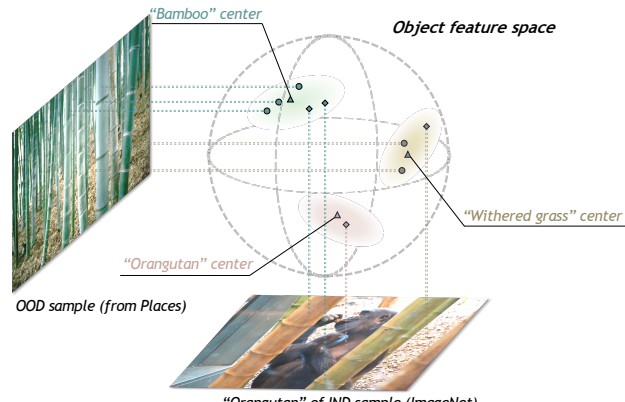

**Figure 1: This toy sample illustrates a challenge in OOD detection using a 3D feature space. The features associated with "Bamboo" and "Withered grass" in the IND image are spatially close to the OOD image's features in this space. This proximity can lead the model to misclassify the IND image, mistaking it for containing similar objects to the OOD image.**

but also raise alerts to humans when encountering unknown samples. OOD detection is the task of determining whether a test sample falls within the in-distribution (IND) or not. This task finds broad applications in autonomous driving [3, 4], fraud detection [37], medical image analysis [12, 41, 51], etc.

Current OOD detection methods [18, 19, 25, 28, 31, 42, 43, 52] often rely on image-level features to construct a score function for identifying OOD samples. In these approaches, a test sample is classified as OOD if its score surpasses a threshold, or vice versa. A major limitation of these approaches is that a single label in the training data fails to fully describe the content of an image, especially when the image contains multiple objects. In such cases, the model can learn "OOD-like" features from the IND data, leading to misclassifications. In essence, the model can learn some "OOD-like" characteristics even without encountering any actual OOD samples during training. A toy example is illustrated in Figure 1. We utilize an image tagging model to generate tags for both an IND image labeled as "orangutan" and an OOD image. Splitting the image based on these tags exposes a critical insight for OOD detection: a part of the IND image shares surprising feature similarity with the OOD image. For example, tags like "bamboo" and "withered grass" might be present in both. This observation highlights a crucial disconnect between the model-captured visual features and the semantic meaning of the image. While the "orangutan" label might be accurate, the presence of "bamboo" and "withered grass" suggests the image includes background elements that could be commonly found in OOD data. This disconnect motivates our exploration of vision-language representations for OOD detection. Our approach

integrates semantic information with visual features and learns the IND area of each image that truly corresponds to the label, resulting in a more refined IND distribution for enhanced OOD detection.

In this paper, we propose **TagOOD**, a novel approach for Out-of-Distribution detection that leverages vision-language representations. TagOOD specifically addresses the challenge of confusing OOD samples, which may contain objects visually similar to objects found in the in-distribution data. These OOD objects are particularly problematic because they not be explicitly labeled within the IND data. TagOOD tackles this challenge through two key mechanisms: 1) decoupling image features using a tagging model to focus on semantic content beyond the object, and 2) generating object-level class centers to capture the central tendencies of IND objects, minimizing the influence of irrelevant background features.

TagOOD leverages a pre-trained tagging model, typically a large vision-language model trained on extensive image-text datasets. This model goes beyond single-label classification by identifying and assigning multiple semantic tags to objects within an image. By incorporating the tagging model, TagOOD focuses on object-level features for OOD detection. Specifically, for a given in-distribution image, the tagging model generates a set of object tags along with their corresponding features. The object features representing IND objects are then fed into a lightweight network for projection into a common feature space. This allows for the subsequent generation of trainable class centers, representing the central tendencies of each object category within the IND data. Finally, TagOOD calculates an OOD score based on the cosine similarity between these learned class centers and a test sample. A high cosine similarity indicates the test sample is likely IND data, as it closely resembles the central tendencies learned from the training data. Conversely, a low cosine similarity suggests the test sample is likely OOD.

The contributions of this paper are summarized as follows:

- Decoupling image features using a vision-language model: allows TagOOD to focus on the semantic content of the object, providing a deeper understanding of the IND data and mitigating the influence of irrelevant background features.
- Generating object-level class centers: by capturing the central tendencies of objects within the in-distribution data, TagOOD can effectively distinguish between IND and OOD samples, even if they contain similar objects.
- To investigate the effectiveness of TagOOD, we conducted comprehensive experiments and ablation studies on the popular ImageNet benchmark. These studies demonstrate competitive performance and reveal the importance of both decoupling features and generating object-level class centers for TagOOD's superior performance.

## 2 RELATED WORK

### 2.1 Vision-Language Models for Image Tagging

Recent advancements [1, 10, 44] in large language models (LLMs) have revolutionized natural language processing (NLP). In computer vision (CV), vision-language models (VLMs) [29, 56] bridge the gap between visual and textual data, leveraging the strengths of both modalities for tasks like image tagging. VLMs fall into two main categories: generation-based and alignment-based. Generation models create captions for images, with recent advancements

in language models [7, 35] making this approach dominant. Unlike earlier methods [9] that relied solely on recognizing tags and then composing captions, these models [5, 26, 50] leverage powerful language models for text generation conditioned on visual information. Alignment models, on the other hand, aim to determine if an image and its description match. Many works [21, 22, 35] typically rely on aligning features from both modalities, often using a dual-encoder or fusion-encoder architecture. Image Tagging, a crucial computer vision task that involves assigning multiple relevant labels to an image, plays a vital role in VLM performance. Traditionally, classifiers and loss functions were used for image tagging. Recent work [30, 39] utilizes transformers and robust loss functions to address challenges like missing data and unbalanced classes. Our work utilizes this powerful VLM technology to enhance OOD detection performance.

### 2.2 Out-of-Distribution Detection

Out-of-distribution detection aims to distinguish OOD samples from IND data. Numerous methods have been proposed for OOD detection. Maximum softmax probability (MSP) [16] serves as a common baseline, utilizing the highest score across all classes as an OOD indicator. ODIN [28] builds upon MSP by introducing input perturbations and adjusting logits through rescaling. Gaussian discriminant analysis (GDA) has also been employed for OOD detection in works by [25, 52]. ReAct [42] leverages rectified activation to mitigate model overconfidence in OOD data. In recent developments, several multi-modal methods [8, 33, 34, 48] have emerged to address the challenge of OOD detection, utilizing the generalized representations acquired through CLIP [11]. CLIPN [48] introduces an extra text encoder to CLIP, aligning entire images with two types of textual information and enabling the model to respond with "no" when faced with OOD samples. MCM [34] aligns image-level features with their corresponding textual description features, enhancing the capabilities of IND estimates and consequently improving OOD detection performance. While CLIP demonstrates remarkable zero-shot OOD detection capabilities, these OOD detection methods utilizing CLIP still neglect to address the challenge of similar object confusion between IND and OOD images. Different from these CLIP-based methods, we attempt to exploit an image tagging model to generate multiple tags in terms of mitigating the impact of confused objects.

## 3 METHOD

### 3.1 Preliminary

Out-of-distribution(OOD) detection aims to distinguish between data the model has seen during training and unseen data from a different distribution. Formally, we can represent the training set as $\mathcal{D}_{IND}^{train} = \{(x_i, y_i)\}_{i=1}^n$, where $x_i \in \mathbb{R}^{3 \times H \times W}$ is the input image typically a 3-channel image with height $H$ and width $W$, $y_i \in \{1, 2, ..., K\}$, one of $K$ possible IND categories, is the corresponding class label, and $n$ is the number of samples in training set $\mathcal{D}_{IND}^{train}$. When a recognition model $F$ is presented with a test set $\mathcal{D}^{test} = \{x_i'\}_{i=1}^p$, the goal is to split the test data into IND and OOD categories, denoted as $\mathcal{D}_{IND}^{test}$ and $\mathcal{D}_{OOD}^{test}$ respectively. Unlike the training data with labeled categories, OOD data typically lacks

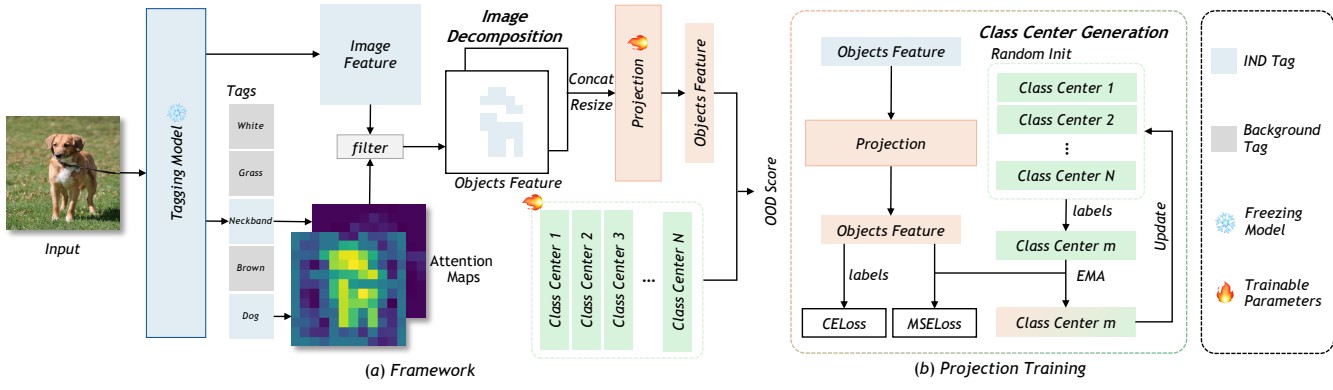

Figure 2: The TagOOD pipeline for OOD detection consists of two main stages, illustrated in (a) and (b). First, image feature decomposition (see (a) on the left) leverages a vision-language model to generate multiple tags. The model then identifies tags belonging to the IND category and creates corresponding attention masks within the image. Next, as shown in (b), the extracted IND object features are used to train a lightweight network that produces a set of IND class centers. During inference (referring back to (a)), TagOOD computes a distance-based metric between IND class centers and test sample features as the OOD score.

labels entirely and does not fall within the set of $K$ possible IND categories. This separation is often achieved by defining an OOD score function $h(x)$ which assigns a score to a test image $x$:

$$h(x) = \begin{cases} 0, & \text{if} \quad x \in \mathscr{D}_{OOD}^{test}, \\ 1, & \text{if} \quad x \in \mathscr{D}_{IND}^{test}, \end{cases} \quad (1)$$

As shown in Equation (1), the function outputs 0 if the image likely belongs to the OOD data, and 1 if likely belongs to the IND data. In practice, the OOD score function provides a continuous value between 0 and 1, with a higher score indicating a higher likelihood of the data point being IND. This score is then used to classify the test data into IND and OOD categories.

## 3.2 Overview of TagOOD

Our proposed TagOOD approach aims to address a key challenge in OOD detection: distinguish the confusing OOD samples, which contain objects similar to objects found in the training data. These objects in test samples can be particularly problematic because they are non-labeled but occur in in-distribution data. Figure 2 illustrates our framework, which consists of two main steps: **1) Image Decomposition.** In Figure 2(a), TagOOD first leverages a pre-trained tagging model to generate multiple semantic tags for each object within an image. The model then identifies tags belonging to the IND category and creates corresponding attention masks within the image. By analyzing the attention maps produced by the tagging model, TagOOD identifies the relevant regions within the image feature extracted by the backbone network of the pre-trained tagging model. This allows our approach to obtain object features for each identified category object(See Sec 3.3 for details). **2) Class Center Generation.** We generate object-level class centers to further accurately construct the IND distribution and eliminate the error impact caused by the tagging model. Once IND object features are extracted, TagOOD employs a lightweight projection model. This model learns to project these object features, which might have different shapes due to the varying nature of objects, into a common feature space. Simultaneously, as Figure 2(b), it updates class centers (see Sec 3.4 for details) for each object category.

During the inference phase, referring back to Figure (a), TagOOD leverages a distance-based metric, such as Euclidean distance or cosine similarity, for OOD detection. This metric directly compares the features extracted from a test sample with the learned class centers representing the IND data. A larger distance or lower cosine similarity between the test features and the nearest class center suggests a higher likelihood of the sample being OOD. This strategy effectively utilizes the learned class centers tha capture the central tendencies of objects within the training data. By identifying deviations from these expected patterns through distance metrics, TagOOD achieves robust OOD detection.

## 3.3 Vision-Language Approach for Image Feature Decomposition

TagOOD leverages a pre-trained vision-language model to identify and label objects within an image, enabling feature decomposition. In practice, we use RAM [56], which combines three components for generating image features, tags, and corresponding attention maps required for feature decomposition: **1) Image Encoder $f$** captures features from the input image, capturing visual information about all of the objects within the image. **2) Text Encoder $g$** based on the CLIP model [38] encodes potential object tags from a pre-defined vocabulary $T$ into a common feature space. This vocabulary $T = \{t_i\}_{i=1}^{M}$, provided by RAM, containing $M$ object used for more detailed image labeling. **3) Tagging Head $\phi$** combines the image features and tag information to predict the final result. The result contains the tags $T^x = \{t\}$ for the objects within the image and corresponding attention maps $A^x = \{a\}$, where $a \in \mathbb{R}^{H \times W}$. In practice, we utilize the last cross-attention module within the Tagging Head to generate attention maps. Eq. 2 summarizes how the tagging model operates:

$$T^x, A^x = \phi(f(x), g(T)), \quad (2)$$

When the vision-language model processes an image $x$, we will obtain an image feature $f(x)$, predicted tags $T^x$ for the objects within the image, and corresponding attention maps $A^x$ for each

predicted tag. We then filter $T^x$ to retain only the tags belonging to our IND vocabulary $T^{in}$.

TagOOD leverages the attention maps $A^x$ to achieve image feature decomposition by selecting IND object features from the full image feature $f(x)$. In Figure 2(a), Our filter applies a threshold $\tau$ to the attention map values corresponding to the object. This essentially creates a binary mask (values above $\tau$ become 1, others become 0), highlighting the most influential image regions of each object. Instead of element-wise multiplication, TagOOD directly selects features from the original image representation corresponding to locations with 1 in the mask. This effectively chooses the parts of the image feature that are most relevant to the objects based on the predicted tag. When multiple IND objects are detected, their corresponding object features are combined into a single feature representation, denoted as $z$. This combined feature vector then undergoes reshaping to prepare it for the next step of the model. Through the process described above, we can obtain a training dataset consisting of object features and their corresponding labels. This dataset is denoted as $\mathscr{Z}_{IND}^{train} = \{(z_i, y_i)\}_{i=1}^n$, where $z_i$ represents the object features for the $i$-th sample, $y_i$ represents the corresponding label for the $i$-th sample, indicating its IND category.

Since the original vocabulary $T$ might not encompass all IND labels, we construct an IND vocabulary which is a subset of $T$, denoted by $T^{in}$. Before training TagOOD, we go through a process to select the most relevant tags $T^{in}$ representing the IND data from $T$. $T^{in}$ are used during training to establish the central tendencies of objects within each category. We collect all the tags associated with the objects within our training data. To ensure the selected tags accurately represent the ground truth, human experts review the results after identifying the most frequent tag within each category. Through this process, we obtain a set of IND tags $T^{in}$ from the predefined vocabulary $T$ that best corresponds to the object categories present in the training data.

## 3.4 Object-Level Class Center Generation

TagOOD employs a lightweight projection model $\boldsymbol{p}$ with dataset $\mathscr{Z}_{IND}^{train}$ to learn a set of object-level IND class centers, denoted as $U = \{\mu^i\}_{i=1}^K$. These class centers represent the typical characteristics of objects belonging to IND categories and serve as effective indicators for OOD detection. Initially, we randomly initialize a set of tensors $\{\mu^i\}_{i=1}^K$ to represent the class centers for each of the $K$ IND categories. During training, as illustrated in Figure 2(b), the model receives the object features $z$, which can have varying sizes depending on the specific objects. The projection model transforms these features $z$ into a common feature space. This ensures consistency when comparing the projected features with $\mu$. The object feature $z^c$ and its label $y^c$ of category $c$ are used to calculate a combined loss function:

$$\mathcal{L} = \alpha \cdot \text{CE}(\boldsymbol{p}(z^c), y^c) + \beta \cdot \text{MSE}(\boldsymbol{p}(z^c), \mu^c), \quad (3)$$

where $CE(\cdot)$ represents the cross-entropy loss, $MSE(\cdot)$ stands for mean squared error loss. The hyperparameters $\alpha$ and $\beta$ control the relative importance of each loss term during training. The loss term $\text{CE}(\boldsymbol{p}(z^c), y^c)$ helps the model learn to differentiate between object features belonging to different IND categories. $\text{MSE}(\boldsymbol{p}(z^c), \mu^c)$ term encourages the model to refine the projected features to better

align with the central tendencies represented by the class centers. Before the next training iteration, the IND class centers are updated. TagOOD utilizes an exponential moving average (EMA) [13] with $\boldsymbol{p}(z)$ to achieve this update. EMA helps smooth out fluctuations in the projected features and provides a more stable estimate of the central tendencies for each IND category. According to the process we described above, the parameter of projection model $\theta^p$ and the class centers $\{\mu^i\}_{i=1}^K$ are updated by the following equations:

$$\theta_{t+1}^p = \theta_t^p - \gamma_1 \frac{\partial \mathcal{L}}{\partial \theta_t^p}, \quad (4)$$

$$\mu_{t+1}^c = (1 - \gamma_2)\mu_t^c + \gamma_2 \boldsymbol{p}(z^c), \quad (5)$$

where $\gamma_1$, and $\gamma_2$ are learning rates controlling the update speed.

## 3.5 OOD Detection Based on the Class Centers

When the training of our projection model $\boldsymbol{p}$ finishes, we will get a set of IND class centers $U$ that capture the central tendencies of the IND object accurately. To assess the likelihood of a test sample $x'$ belonging to the IND data, we compute the cosine similarity between its projected features $z'$ and each class center within $U$. The maximum cosine similarity score across $U$ is used as the OOD score for the test sample, denoted by $\boldsymbol{h}(x')$. Intuitively, a high cosine similarity score indicates that the test sample feature closely resembles one of the learned IND class centers $\mu^i$, suggesting it's likely IND data. Conversely, a low OOD score suggests the test sample deviates significantly from the expected IND patterns. The provided mathematical formula accurately represents this calculation:

$$\boldsymbol{h}(x') = \max_c \left( \frac{z' \cdot \mu^c}{\|z'\|\|\mu^c\|} \right). \quad (6)$$

If the tagging model predicts no tags in $T^{in}$ for a given image, TagOOD directly classifies the sample as OOD. Overall, by comparing the test sample feature ($\boldsymbol{p}$ supports both image feature or object feature) to the learned class centers using cosine similarity, TagOOD can effectively distinguish between in-distribution and out-of-distribution data, enhancing the model's robustness in real-world applications.

## 4 EXPERIMENTS

### 4.1 Experimental Setup

**In-distribtuion Dataset.** We adopt the well-established ImageNet-1K [40] dataset, a standard dataset for OOD detection, as our in-distribution data. ImageNet-1K is a large-scale visual recognition dataset containing 1000 object categories and 1281167 images.
**Out-of-Distribution Dataset.** Based on ImageNet as IND datasets, we follow Huang *et al.* [20] to test our TagOOD with iNaturalist [46], SUN [53], Places365 [57], and Texture [6]. To further explore the generalization ability of our approach, we follow Wang *et al.* [49] and employ another two OOD datasets, OpenImage-O [24] and ImageNet-O [17] to evaluate our method.
**Evaluation Metrics.** We evaluate our approach using common OOD detection metrics [20, 34, 42, 54, 55]: **Area Under the ROC Curve (AUROC)**. This metric summarizes the model's ability to distinguish between IND and OOD samples. Higher AUROC indicates better overall performance. **False Positive Rate at 95% True Positive Rate (FPR95)**. This metric measures the model's

**Table 1: OOD detection performance comparison of TagOOD and existing methods. The best and second-best results are indicated in Bold and Underline. The method marked with * indicates that it utilizes a more expansive backbone in comparison to the others. ↑ indicates larger values are better and ↓ indicates the opposite. All values are expressed in percentages.**

| Method | OOD Datasets | | | | | | | | Average | |
| | iNatrualist | | SUN | | Place | | Texture | | | |
| | AUROC↑ | FPR95↓ | AUROC↑ | FPR95↓ | AUROC↑ | FPR95↓ | AUROC↑ | FPR95↓ | AUROC↑ | FPR95↓ |
|---|---|---|---|---|---|---|---|---|---|---|
| MSP[16] | 95.60 | 18.89 | 86.88 | 49.18 | 85.64 | 51.73 | 85.17 | 49.66 | 88.32 | 42.37 |
| ODIN[28] | 71.04 | 48.53 | 60.47 | 69.00 | 57.01 | 72.69 | 63.03 | 68.88 | 62.89 | 64.78 |
| ENERGY[31] | 94.87 | 14.47 | 83.44 | 45.34 | 82.25 | 48.55 | 80.64 | 52.93 | 85.30 | 40.32 |
| ReAct[42] | 97.61 | 7.32 | 86.71 | 40.30 | 85.14 | 44.49 | 82.99 | 48.97 | 88.11 | 35.27 |
| MCM(CLIP-L)*[34] | 94.95 | 28.38 | **94.12** | **29.00** | **92.00** | **35.42** | 84.88 | 59.88 | 91.49 | 38.17 |
| LHAct[55] | 63.60 | 58.53 | 65.10 | 92.11 | 64.84 | 92.69 | 81.06 | 85.35 | 68.65 | 82.17 |
| FeatureNorm[54] | 66.42 | 89.70 | 60.77 | 92.22 | 61.44 | 91.80 | 55.86 | 95.78 | 61.12 | 92.38 |
| TagOOD(Ours) | **98.97** | **5.00** | 92.22 | 29.70 | 87.81 | 40.40 | **90.60** | **36.31** | **92.40** | **27.85** |

**Table 2: Evaluation on more challenging datasets. The method marked with * indicates that it utilizes a more expansive backbone in comparison to the others. The best and second-best results are indicated in Bold and Underline. ↑ indicates larger values are better and ↓ indicates the opposite. All values are expressed in percentages.**

| Method | OOD Datasets | | | | Average | |
| | ImageNet-O | | OpenImage | | | |
| | AUROC↑ | FPR95↓ | AUROC↑ | FPR95↓ | AUROC↑ | FPR95↓ |
|---|---|---|---|---|---|---|
| MSP[16] | 85.22 | 59.25 | 92.57 | 29.89 | 88.90 | 44.57 |
| ODIN[28] | 72.85 | 57.15 | 69.21 | 52.73 | 71.03 | 54.94 |
| ENERGY[31] | 87.33 | 37.05 | 86.64 | 36.83 | 86.99 | 36.94 |
| React[42] | **87.85** | **34.45** | 93.39 | 20.81 | 90.62 | **27.63** |
| MCM(CLIP-L)*[34] | 82.46 | 68.75 | 92.92 | 35.84 | 87.69 | 52.30 |
| LHAct[55] | 57.72 | 91.75 | 59.27 | 70.66 | 58.50 | 81.21 |
| FeatureNorm[54] | 54.22 | 94.84 | 61.00 | 92.83 | 57.61 | 93.84 |
| TagOOD(Ours) | 87.48 | 51.91 | **96.28** | **20.44** | **91.88** | 36.18 |

tendency to misclassify OOD samples as IND samples when the true positive rate (correctly identified IND samples) is 95%. Lower FPR95 suggests the model effectively identifies OOD data even with a high true positive rate for IND data.

**Training Details.** We leverage RAM [56] as the underlying vision-language model for OOD sample detection. For consistency across all compared methods, the feature extractor within RAM, swin-large [32], is used for all approaches. Our lightweight projection model employs two serially connected self-attention block layers. The projected feature space has a dimensionality of 512. ImageNet-1k is used for training, with a total of 100 epochs, and the class centers are initialized with Gaussian noise for the 1000 ImageNet categories. The hyperparameters $\alpha$, $\beta$, $\gamma_1$, $\gamma_2$ are set to 1, 0.1, 1, $1 \times 10^{-4}$, respectively. These values were chosen based on our experimental results. During training, The Adam optimizer [23] is used during training with an initial learning rate of 0.01, a Cosine Annealing learning rate scheduler, and a batch size of 256. All the experiments are performed using PyTorch [36] with default parameters on two NVIDIA V100 GPUs.

## 4.2 Main results

**Standard evaluation on ImageNet.** We compare the performance of our TagOOD against seven popular OOD detection approaches, including MSP [16], ODIN [28], Energy [31], ReAct [42], MCM [34], LHAct [55], FeatureNorm [54]. All methods, except MCM, utilize Swin-L as the backbone extractor for a fair comparison. MCM employs a larger and more complex CLIP-based ViT-L model. As shown in Table 1, our method achieves better AUROC and FPR95 metrics. We highlight that TagOOD achieves 27.85% on FPR95, which outperforms the previous best method ReAct [42] by 7.42% across various OOD datasets. While MCM [34] exhibits higher AUROC and lower FPR95 on SUN and Places datasets due to its significantly larger **CLIP ViT-L backbone**, which is 1.7 times the size of Swin-L, TagOOD maintains the best average performance across all datasets. Our analysis suggests that LHAct [55] and FeatureNorm [54] might suffer limitations due to hyperparameter settings optimized for a different backbone network. This mismatch between hyperparameters and the chosen Swin-L model can lead to suboptimal performance on these methods.

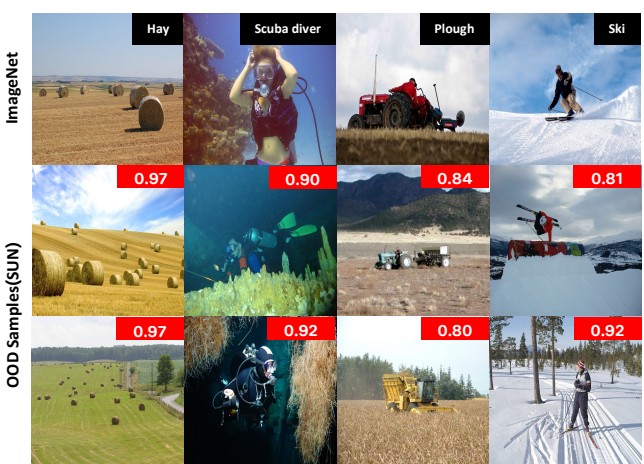

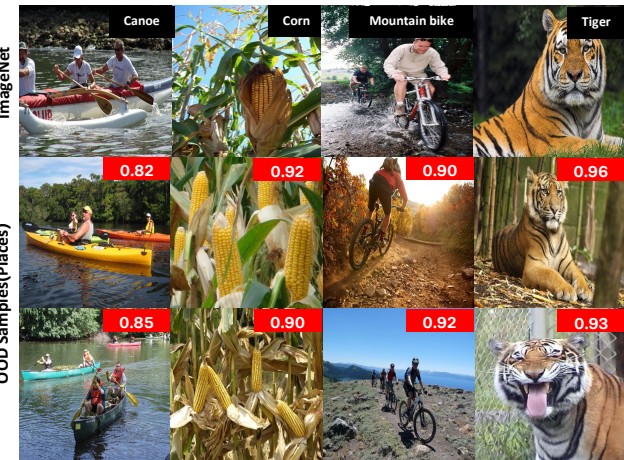

Figure 3: Overlap between IND and OOD samples. the black-tagged images are from ImageNet for reference and represent a standard IND class sample. The samples with red tags containing IND class objects are selected from the SUN and Places datasets. Red tags indicate the corresponding OOD scores assigned by our method.

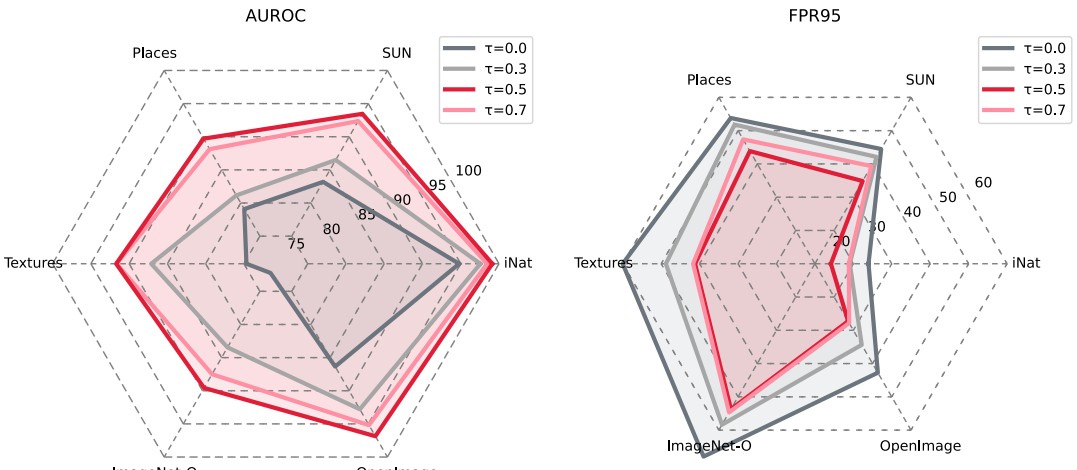

Figure 4: The performance of TagOOD training with varying values of the hyperparameter $\tau$.

The study by Bitterwolf *et al.* [2] highlights a crucial aspect of OOD detection: the overlap between training and OOD data. The commonly used datasets in our evaluation (detailed in Section 4.1) exhibit varying degrees of overlap with the ImageNet-1k training data. This overlap can potentially influence the evaluation of a model to identify OOD samples. In Figure 3, the black-tagged images are from ImageNet for reference and represent a true IND class sample. The samples with red tags, selected from the SUN and Places datasets, contain IND class objects. These red tags indicate the corresponding OOD scores assigned by our method. TagOOD correctly assigns them OOD scores with high confidence, indicating it is reliable to accurately distinguish IND objects. This highlights the robust OOD detection performance of TagOOD, leveraging the power of the vision-language model.

**Evaluation on more challenging OOD datasets.** To further assess the limitations and robustness of our TagOOD, we conducted experiments on two challenging datasets: OpenImage-O [24] and

ImageNet-O [17]. OpenImage-O contains a diverse set of OOD samples, while ImageNet-O specifically includes adversarial examples designed to fool detection models. As shown in Table 2, TagOOD demonstrates strong performance on OpenImage-O, indicating its effectiveness in handling general OOD data. However, on ImageNet-O, which contains specifically crafted adversarial examples, the performance of TagOOD shows some limitations while still achieving the best AUROC score. This suggests that further exploration is needed to enhance TagOOD's resilience against adversarial attacks.

### 4.3 A closer look of TagOOD

**How does image feature decomposition affect the performance?** We investigate the influence of the hyperparameter $\tau$ on image feature decomposition for our projection model training. Figure 4 visualizes the effects of varying $\tau$ values on model performance. Specifically, we set $\tau = \{0, 0.3, 0.5, 0.7\}$. Higher $\tau$ values correspond to a more selective feature subset, prioritizing those

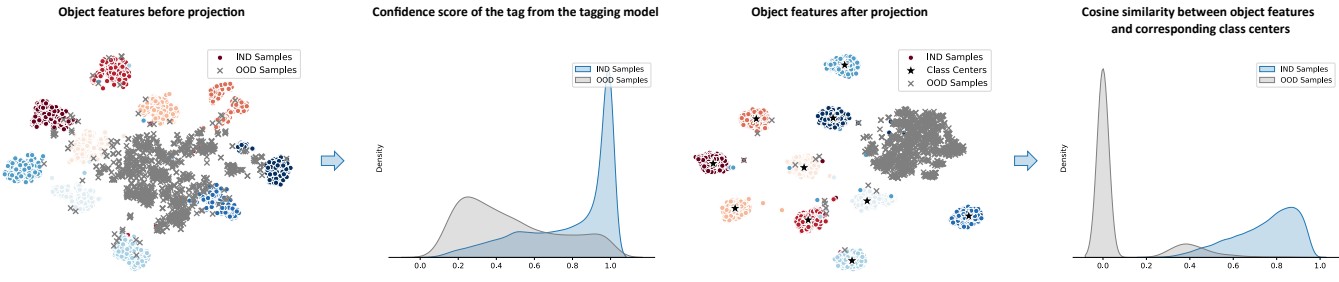

**Figure 5: Illustration of object features visualization and OOD score distribution. Both object features are visualized by T-SNE [45]. Data points represent object features, and colors encode their corresponding IND class labels. Gray X marks indicate OOD data points. The features presented on the left are directly extracted from the tagging model before projection. Following the projection process, the object features become more condensed, as demonstrated on the right.**

with strong alignment to IND object classes. Conversely, $\tau = 0$ represents a baseline where all image features are directly used for object model training. We observe that the optimal performance across all the OOD datasets used in our evaluation is achieved at $\tau = 0.5$. This finding suggests a critical balance between feature selectivity and information retention for effective OOD detection. Interestingly, the performance suffers more significantly when $\tau$ decreases from 0.5. In contrast, increasing $\tau$ beyond 0.5 leads to only a slight performance degradation. We hypothesize that this occurs because higher $\tau$ values eliminate some redundant information without discarding critically important features for accurate OOD classification. In essence, our model demonstrates the ability to retain the most crucial information from image samples while discarding irrelevant features at $\tau = 0.5$. This selectivity contributes to the reliable performance of TagOOD in identifying OOD data.

**Table 3: A set of ablation results for TagOOD, averaged across four standard OOD datasets. CCG stands for class center generation, Proj. represents the projection model training, CS means Cosine Similarity distance and CE and MSE represent Cross Entropy loss and Mean Square Error loss.**

|  | CCG | Proj. | AUROC↑ | FPR95↓ |
|---|---|---|---|---|
| Tag Score |  |  | 85.62 | 43.44 |
| CS | √ |  | 83.28 | 39.94 |
| $p$(CE)+CS |  | √ | 91.58 | 32.13 |
| $p$(CE+MSE)+CS | √ | √ | **92.40** | **27.85** |

**What is the effect of the projection model and class centers?** Table 3 and Figure 5 systematically explore the influence of the projection model and class centers on the performance of TagOOD. This analysis helps us understand how each step contributes to the final OOD detection effectiveness. Tag Score in Table 3 utilizes the confidence score directly from the tagging model as the OOD score. CS leverages object features obtained after the image feature decomposition. In this case, we compute the average feature of each IND class as the class center. Cosine similarity is then computed as the OOD score. $p$(CE)+CS uses the projection model trained solely with a CE loss to project the object features. Subsequent operations are the same as CS. The bottom line represents the full

TagOOD approach. It incorporates all the previous steps: image feature decomposition, a projection model trained with a combined loss function which includes both CE loss and MSE loss, and class center generation. As observed, each stage progressively improves performance, culminating in superior OOD detection capabilities. This highlights the importance of both the projection model and class centers in achieving reliable OOD detection.

To gain deeper insights into the operation of TagOOD, Figure 5 utilizes T-SNE [45] for visualization. The data points represent object features, and colors encode their corresponding IND class labels. The gray 'X' markers represent OOD samples. A critical observation is the improved feature separation achieved through projection. Features that are directly procured from the tagging model, shown on the left, are likely to be scattered and overlapping. On the right side, we see the features after the projection operation in TagOOD. These features form tighter and more distinct clusters, with clearer boundaries between different IND classes. This enhanced separation in the projected feature space allows TagOOD to assign more distinguishable OOD scores. This facilitates the identification of OOD samples because their features deviate significantly from the learned representations of known object classes.

**Table 4: Performance of TagOOD using various distance metrics. Results are averaged across four standard OOD datasets.**

| Metrics | AUROC↑ | FPR95↓ |
|---|---|---|
| KL Divergence | 90.66 | 32.53 |
| Euclidean Distance | 92.03 | 28.74 |
| **Cosine Similarity** | **92.40** | **27.85** |

**Is TagOOD robust with various distance metrics?** This ablation experiment investigates the sensitivity of TagOOD to various distance metrics commonly used in OOD detection: KL divergence, Euclidean distance, and cosine similarity. The results in Table 4 demonstrate that TagOOD exhibits robustness across these metrics, maintaining strong performance. While KL divergence and Euclidean distance show slight performance degradation compared to cosine similarity, The effectiveness remains consistent. This suggests that TagOOD is not overly reliant on a specific distance metric for accurate OOD detection.

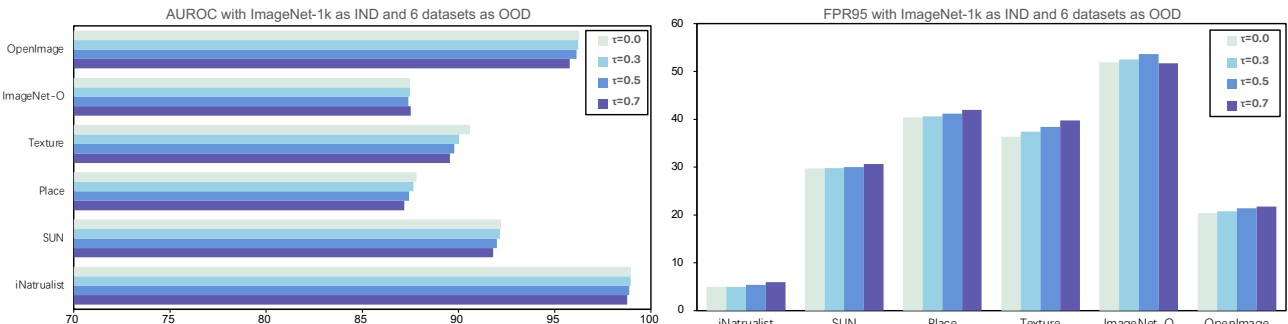

**Figure 6: Results of evaluation on varying levels of features selected during image feature decomposition.**

**Is the performance of TagOOD enhanced when training utilizes a single ground truth tag as opposed to multiple predicted tags?** This ablation experiment explores the influence of tagging strategies on object feature selection for OOD detection. We evaluate two strategies: utilize a single tag corresponding to the ground truth for feature decomposition or leverage the predictions of the tagging model, generating multiple tags for each image. The decomposition is then guided by the attention masks constructed based on these predicted tags. As shown in Table 5, the multiple tags strategy achieves superior performance compared to using the single ground-truth tag. We hypothesize that relying solely on the ground truth might lead to inaccurate feature decomposition due to potential tagging model errors. These errors could introduce "missing information" or "offset issues" in the decomposed features, hindering the ability to capture essential characteristics for effective OOD detection. In contrast, the multiple tags strategy incorporates the predictions of the tagging model, potentially offering a richer representation of the image content. This can lead to more accurate feature decomposition and the extraction of more informative features, ultimately enhancing OOD detection capabilities.

**Table 5: Ablation experiment about impact of tag selection strategies on TagOOD Performance.**

| Tag for Training | AUROC↑ | FPR95↓ |
|---|---|---|
| One tag | 91.71 | 31.41 |
| **Multiple tags** | **92.40** | **27.85** |

**Does TagOOD Maintain Robustness with Feature Variations?** This section investigates the ability of TagOOD to distinguish OOD data at inference despite variations introduced (different values of $\tau$) during image feature decomposition. As mentioned earlier, our projection model utilizes both image and object features. To explore the robustness of TagOOD against such variations, we evaluate TagOOD on all six OOD datasets while adjusting the hyperparameter $\tau$ in the image feature decomposition stage. This manipulation effectively controls the level of detail extracted from image features. The results visualized in Figure 6 are encouraging. They demonstrate that TagOOD consistently maintains strong performance in OOD detection, regardless of the utilization of either image features or object features, and irrespective of how the hyperparameter $\tau$ impacts

image feature decomposition. This suggests that the projection model in TagOOD effectively learns discriminative features and exhibits robustness to variations in feature representation extracted during image feature decomposition.

## 5 LIMITATIONS

While TagOOD demonstrates promising results, there are areas where further exploration could enhance its capabilities. The performance of the tagging model can influence TagOOD's ability to accurately detect OOD data. Currently, TagOOD solely leverages the backbone network of the tagging model for feature extraction. This limits the potential for exploring alternative feature extractors that might be better suited for OOD detection tasks. Opening doors for investigating alternative feature extraction techniques will be an exciting future direction for this research.

## 6 CONCLUSION

Existing OOD detection methods often rely on whole-image features, which can incorporate irrelevant information beyond IND objects. This paper introduces TagOOD, a novel approach that leverages the power of vision-language models (VLMs) to achieve more focused OOD detection. TagOOD tackles the challenge of OOD detection by decoupling images from their corresponding classes and generating class centers within a common feature space. This innovative approach offers many advantages: TagOOD employs a VLM to decouple images from their associated classes. This allows the model to focus on the essential semantic information and learn the IND area of each image that truly corresponds to the label, resulting in a more refined IND distribution for enhanced OOD detection. TagOOD generates object-level class centers for each category within the IND data. These class centers serve as reference points in the common feature space, enabling the model to distinguish OOD samples effectively. A distance-based metric is used between the test sample feature and class centers for OOD detection. This simple yet effective approach allows the model to identify samples that fall outside the expected distribution of the IND data. This work presents a novel perspective for further exploration of multimodal information utilization in OOD detection, we hope our work contributes to the advancement of robust and reliable AI systems.

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
