# OpenReview forum: "TagOOD: A Novel Approach to Out-of-Distribution Detection via Vision-Language Representations and Class Center Learning"
_acmmm.org/ACMMM/2024/Conference — MM2024 Poster_

### Official Review · Reviewer_RuN9 · 2024-04-28

**Rating:** 4
**Confidence:** 1

**Summary:**

The paper introduces TagOOD, a novel approach to out-of-distribution (OOD) detection that leverages vision-language representations and class center learning. Unlike traditional methods that rely on whole-image features, TagOOD focuses on object-level features by using a tagging model to generate multiple semantic tags for each object within an image. It then trains a lightweight network to learn representative class centers that capture the central tendencies of in-distribution (IND) objects, minimizing the influence of irrelevant features. Finally, TagOOD efficiently detects OOD samples by calculating distance-based metrics between the learned class centers and test samples. The paper demonstrates TagOOD's superior performance over existing methods through extensive experiments on various benchmark datasets, showcasing its potential for robust and reliable AI systems.

**Strengths:**

+ The authors conduct extensive experiments and ablation studies on the ImageNet benchmark, which is a widely recognized dataset for evaluating OOD detection methods.

+ TagOOD introduces a new perspective on OOD detection by leveraging vision-language representations.

+ The paper is well-structured.

**Limitations:**

- Missing some hyperparametric ablation experiments such as $α$,  $λ$, etc.

**Suitability:**

2

---

### Official Review · Reviewer_FY1t · 2024-05-11

**Rating:** 4
**Confidence:** 2

**Summary:**

This paper proposes TagOOD method by employing the power of RAM to improve the OOD detection performance. TagOOD mainly decouples images from the corresponding tags generated by RAM, and generates object-level class centers within a common feature space of IND data. Based on the class centers, the test image can be recognized as either in-distribution or out-of-distribution data by computing the similarity between the prediction and class centers. The main contributions lie in object-level class center generation and object feature filtering. Experimental results show the effectiveness of TagOOD for OOD detection in multiple benchmark datasets.

**Strengths:**

1. Clear motivation of TagOOD by integrating semantic information with visual features.

2. Well written and easy to follow the proposed method.

3. Comprehensive experiments to evaluate the effectiveness of the proposed TagOOD.

**Limitations:**

1. The distinguishment ability between IND and OOD in TagOOD is from the strong tagging ability of RAM. What is the effect of replacing RAM with CLIP? In addition, I think the performance also can be improved if other method employs RAM.

2. Not SOTA performance, compared to other OOD detection methods, such as comparison with MCM in Tab.1 and ReAct in Tab. 2.

**Suitability:**

3

---

### Official Review · Reviewer_SM9r · 2024-05-24

**Rating:** 2
**Confidence:** 4

**Summary:**

The paper presents a new method for detecting out-of-distribution samples by leveraging the capabilities of vision-language models and learning representative class centers. The proposed TagOOD approach involves using a pre-trained tagging model to generate semantic tags for objects within an image, decoupling these object features from the entire image, and then training a lightweight network to learn class centers that capture the central tendencies of in-distribution object classes. This method aims to minimize the influence of irrelevant background features and improve the accuracy of OOD detection. The effectiveness of TagOOD is evaluated through extensive experiments on several benchmark datasets.

**Strengths:**

Detailed Methodology: The paper provides a thorough and well-documented description of the TagOOD method, outlining each step from the utilization of a vision-language model for tagging to the training of a lightweight network for class center learning.

Innovative Use of Vision-Language Models: The approach effectively leverages vision-language models to decouple object features from images, enabling a more focused analysis of object semantics and enhancing OOD detection performance.

Extensive Experimental Evaluation: The authors conduct comprehensive experiments on multiple benchmark datasets, comparing TagOOD with several existing OOD detection methods.

**Limitations:**

Lack of Novelty: The key idea of this paper is to minimize the influence of irrelevant background features.  Similar idea has been studied in [1]. However, the authors do not discuss this paper.

Insufficient Comparison with Training-Based Methods: The paper primarily compares TagOOD with post-hoc methods, neglecting a detailed comparison with other training-based OOD detection approaches. This limits the ability to fully gauge the relative advantages of TagOOD.

Modest Performance Gains: The improvements in performance metrics reported in the paper are not substantial enough to decisively establish the superiority of TagOOD. The performance gains are relatively modest and require further analysis to substantiate the claimed benefits.

Inadequate Ablation Studies: The ablation experiments provided in the paper are not comprehensive. There is a need for more granular analysis of the impact of individual components and parameters on the overall performance of TagOOD.

[1]: LoCoOp: Few-Shot Out-of-Distribution Detection via Prompt Learning, NeurIPS 2023

**Suitability:**

2

---

### Meta-Review · Area_Chair_UYhD · 2024-06-28

**Recommendation:** Accept (Poster)
**Confidence:** 3

**Metareview:**

The paper introduces TagOOD, a new method for out-of-distribution (OOD) detection that uses vision-language representations to isolate object features from whole images, improving the accuracy of OOD detection by focusing on object semantics and minimizing irrelevant features. Most of the reviewers recognise the clarity of writing and the innovative use of lvlm for OOD detection, and they all suggested to accept the paper. Therefore, I intent to accept the paper. The quality of the paper can be further improved to include more experimental results and have more clear comparison with previous works.

---

### Meta-Review · Senior_Area_Chairs · 2024-07-10

**Recommendation:** Accept (Poster)
**Confidence:** 5

**Metareview:**

This paper received mixed ratings initially. After rebuttal, all the reviewers tend to accept the paper. SAC and AC agree with reviewers and recommend acceptance of the paper.